# Microstructural Surface Properties of Drifting Seeds—A Model for Non-Toxic Antifouling Solutions

**DOI:** 10.3390/biomimetics4020037

**Published:** 2019-05-13

**Authors:** Antje Clasen, Antonia B. Kesel

**Affiliations:** 1Institute of Environmental and Biotechnology, City University of Applied Sciences, 28199 Bremen, Germany; 2Biomimetics-Innovation-Center, City University of Applied Sciences, 28199 Bremen, Germany; akesel@bionik.hs-bremen.de

**Keywords:** biofouling, drifting seeds, surface structure, biomimetic, technical surface, SEM, biofouling analysis

## Abstract

A major challenge in the shipping and marine industry is the biofouling on under water surfaces. So far, biocides have been the main remedy for the prevention of the adhesion of microorganisms that is also influenced by surface topography. In recent years, research projects have explored microstructured surfaces as a non-toxic antifouling strategy. In this study, physical factors of surfaces of seeds of 43 plant species were analyzed with regards to their antifouling effects. After exposure to cold water of the North Sea during the swarming periods of the barnacles larvae, the surface microstructures of seeds without fouling of barnacles were identified and compared with each other, using a scanning electron microscope (SEM). In order to validate the findings, selected microstructured surface structure properties were transferred to technical surfaces with a 2-component silicon system and subjected to the same conditions. The results of the analyses confirmed that drifting seeds with specific microstructural surface structure properties promote biofouling defense of epibionts. These results serve as a starting point for the development of non-toxic antifouling agents based on the interaction of microstructures and geometric shapes.

## 1. Introduction

Biofilms are ubiquitously distributed in natural and industrial environments and play an important role in our global ecology, as 99% of microorganisms live in biofilms [1,2]. Microorganisms do not live as pure cultures of dispersed single cells but accumulate at interfaces to form polymicrobial aggregates under optimal conditions [3]. They colonize almost all surfaces as soon as sufficient nutrients and adequate moisture are present. The undesirable formation of biofilms is called biofouling—a hierarchical and highly dynamic process in which specific organisms develop a fouling community or even an epibiosis depending on substratum, geographical location, seasons and other factors (i.e., competitions and predation) [4].

The adhesion of these microorganisms is influenced, among other things, by their adhesion mechanism and their surface topography [5]. In a next stage, microorganisms serve as a food basis for larvae of macrofoulers (e.g., mussels, barnacles), which settle and develop into adult organisms within a few weeks.

The biofouling leads to major problems in the maritime and shipping industries. Aquatic microorganisms, including fungi, protozoa, algae, and invertebrates, accumulate on the surfaces, particularly on those of seagoing vessels, buoys, sonar equipment, pontoons, offshore constructions, underwater instruments and seawater cooling systems [6]. Considerable financial losses result from the damage of materials (e.g., corrosion), the decrease in hull strength or even more drastically breaches in the hull structure, the weight increase, the increase of the flow resistance and/or the resulting loss of speed and range, the increase of fuel consumption to compensate the same and the resulting increase of the pollutant emissions. In total, the industry is looking at expenditures of approximately €120 billion per year for financing maintenance and countermeasures [7,8,9,10]. In addition, the invasion of foreign species is also encouraged, which leads to an impairment of the ecological balance in the marine ecosystem.

In order to counteract biofouling and its resulting problems, Phoenician ships were already fitted with lead-coated plates [11]. Later, copper strips were nailed to the wooden outer skin of the ships or coatings of zinc metals were used. In the course of time, many differently effective toxic antifouling agents were developed. In 1975, a breakthrough was finally achieved with the use of organotin-based paints, which provide continuous protection against fouling for up to seven years [12]. One of the best-known and most effective organotin-based antifouling agents is biocide tributyltin (TBT). It contains oxidative phosphorylation, thus preventing the biochemical process of adenosintriphosphat (ATP) synthesis in the mitochondria of animals or photophosphorylation in the chloroplasts of plant fouling [13] and drastically interfering with the regulation process of the cells. However, the leaching of TBT into water is not only damaging organisms directly attached to the ships, but also other non-target organisms. Studies showed that imposex are affected by more than 200 gastropod species worldwide [14]. The strongly negative influences of TBT on the fauna gradually led to a rethinking in society and to legal restrictions for the use of TBT in antifouling paints. In many Western European countries, TBT was regulated in antifouling paints in the 1980s. In 2003, the International Maritime Organization (IMO) issued a worldwide ban on the use of paints containing organotin. A total ban on TBT-containing paints followed in 2008 [14,15,16,17].

Currently, most alternative antifouling paints are based on copper compounds such as copper I oxide and copper thiocyanate with a supplementation of biocides such as Irgarol, diuron, zinc pyrithione, copper pyrithione and chlorothalorin. Eighteen different biocides are still in use. Alternative paints are less effective and less stable. However, even these so-called “biocide-free” antifouling agents are toxic to aquatic organisms [18,19]. Therefore, microstructured surfaces as potential non-toxic antifouling strategies for the substitution of biocides have gained much attention in recent years. Nature already provides many solutions for the mimicry of microstructured surfaces. Organisms have adapted to their ecological niches so that they can fend off epibionts in their habitat successfully. Thus, they show a variety of different surface structures such as grooves, dents, spines or scales as well as surface sizes as antifouling mechanisms. Studies of pilot whale and shark hides as well as shells of mussels and crab housings have shown promising results [20,21].

So far, research focused on the antifouling mechanisms of marine organisms. However, the source material of forms in nature, that could serve as biological models, is far from having been exhaustedly researched. Only 1.8 million species have been catalogued to date—a very small part compared to the 8–15 million species that are estimated to exist worldwide [22]. Among others, plants are a major group that has received little attention so far. Research encompasses around 370.000 terrestrial plants [22] and concentrates mainly on identification, isolation and effects of plant ingredients, on surfaces of plant leaves in terms of structure, morphology and properties such as superhydrophobicity, self-cleaning, adhesion and antifouling. Seeds of land plants and their defense against biofouling have been examined in detail only on few occasions. For this reason, plant seeds of tropical and subtropical origins found near the coast constitute the focus of this study. They serve as a model for biocide-free antifouling coatings. None other than Charles Darwin already investigated the drifting of seeds and fruits across the oceans on his expedition with the HMS Beagle in the 19th^.^ century [23]. These drifters can float on the surface for up to 14 months depending on earth movement, ocean current, friction between wind and surface water, salinity and temperature [24]. One of the prerequisites for the long floating period is that they remain free of vegetation. The present study focuses on the extent to which physical properties or surface structures of the seed capsules are responsible for preventing fouling. The surface properties of the seed are identified using scanning electron microscopic and photo-optical measurement methods, transferred to technical surfaces and then subjected to growth analysis. It uses technical surfaces modeled on the drifting seeds for the experiments. The influence of the chemical ingredients of the seeds has been explicitly disregarded for this study in order to be able to focus solely on the surface properties and to avoid additional reducing effect on vegetation that are triggered by leaching of the chemical components. Furthermore, a complete identification and the investigation of the effectiveness of the seed ingredients are very complex and require a sophisticated research setting that cannot be implemented thus far.

## 2. Materials and Methods

### 2.1. Plant Material

For the purpose of this study, seeds of different species were purchased from Sunshine Seeds (Ahlen, Germany) and from Rarepalmseeds.com (Munich, Germany). These seeds originate from plants of different plant families in various regions (Table 1). All seeds were harvested in the country of origin, freshly picked in season ensuring the maturity, and sent with proper and professional storage.

Before the start of the experiments, length and width of all seeds were measured, using a digital caliper (PROMAT, Germany).

### 2.2. Surface Analysis of Drifting Seeds with Scanning Electron Microscopy (SEM)

The surfaces of the drifting seeds were identified by scanning electron microscopy. Selected air dried drifting seeds were mounted on holders using conductive double-sided adhesive tab stripes (Plano, Wetzlar, Germany) and sputter-coated with gold at 60 mA for 30 s (Baltec SCD 005, BAL-TEC, Witten/Ruhr, Germany). Samples were examined in the scanning electron microscope JSM-6510 (JEOL, Tokyo, Japan) at an accelerating voltage between 5–15 kV.

### 2.3. Quantification of the Three Dimensional Surface Roughness of Drifting Seeds

The surface of the drifting seeds and the reference glass balls with a diameter of 5 mm (Silibeads Type M, Sigmund Lindner GmbH, Warmensteinach, Germany) were analyzed employing three dimensional topographical measurements using a stereo-scanning electron microscope (Vega-Serie, TESCAN GmbH, Dortmund, Germany) with an eucentric tilting sample stage at an accelerating voltage of 15 kV For the analysis, the selected air dried drifting seeds were sputtered with gold for 150 s (Baltec SCD 050). The obtained SEM images (tilt angles 0°–10°–20°, 5 measurements per surface and tilt angles, at 400× magnification) were transformed into a digital elevation model (DEM) and the mean roughness (Ra, mean deviation of all measuring points from mean height in the roughness profile) were determined by means of Mex Metrology Version 5.1 (Alicona Imaging GmbH, Schönau, Germany), a software which is designed for surface metrology applications and measures areal parameters based on ISO standards such as 4287/4288 [24].

### 2.4. Reproduction of the Surface Structure of the Drifting Seeds on Technical Surfaces

The surface structures of the drifting seed species identified by SEM were transferred to technical surfaces. These were manufactured with the 2-component silicone Elastosil^®^ M4601 system (R&G Faserverbundwerkstoffe GmbH, Waldenbruch, Germany), its components A (base material) and B (hardeners) thoroughly mixed in a ratio of 9:1. This system is characterized by a tear strength of 6.5 N/mm^2^ (ISO 37), a tear resistance > 30 N/mm (ASTM D 624 B), a density of 1.13 g/cm^3^ at 23 °C in water (ISO 2781) and a hardness of shore A 28 (ISO 868). In order to create specific surface properties, the silicone mass was enriched either with hollow glass particles (Dennert Poraver GmbH, Schlüsselfeld, Germany) or with Q-Cel lightweight hollow microspheres (Omega Minerals GmbH, Norderstedt, Germany), weighing 8% of the silicone mass.

This mixture was poured onto a transparent conventional 15 × 15 cm plexiglass plate (Hornbach, Germany), which has been coated with a bonding agent (base coat G 795, R&G Faserverbundwerkstoffe GmbH) and provided with 2 holes for the hanger. The silicone mass was then evenly distributed on the plate with the help of a paint roller and dried for 20 h at room temperature.

These technical surfaces were analyzed before and after the exposition with the help of the SEM (see Section 2.2 and Section 2.6). In addition, a V2A steel surface was used as reference indicator for showing the potential amount of biofouling in the area.

### 2.5. Fouling Experiments with Drifting Seeds

Experiments were carried out in the North Sea in a sheltered bay near Meldorf (Germany, 54°5′31.45″ N, 8°57′20.05″ E). The area has a tidal range of 3.20 m with an average temperature of 2.0 °C from December to February and 16.7 °C from June to August.

The seeds of all selected species (Table 1) were exposed to the sea water at the study side to investigate their general antifouling properties. Five seeds of each species were placed in one net which was crafted of green plastic mesh with a square base 11 × 50 cm and a mesh size of 10 mm. Additional nets were crafted with gauze (1.1 × 1.2 mm mesh size) to prevent the seeds of slipping through. Samples were exposed in a depth of 0.20 m for a time span of 12 weeks each during the swarming period of the barnacles larvae. The fouling nets were collected in monthly intervals to control their conditions.

After the 12-week exposure, the seeds were removed from the nets and briefly rinsed in freshwater to remove the loose foulness that was not relevant for this study. They were dried at room temperature for 7 days. The subsequent growth analysis focused on the settlement of barnacles and documented each seed photographically.

Seeds which were free of fouling were subjected to additional expositions in three time periods, while barnacle larvae (e.g., *Balanus improvisus*) were present in the sea water. After the growth analysis of the selected seeds, each seed was documented photographically and its dry weight was determined.

### 2.6. Fouling Experiments with Technical Surface According to the Model of the Drifting Seeds

The technical surfaces (see Section 2.4) were exposed to the cold water in the North Sea before they were inspected and subjected to a fouling analysis analog to the drifting seeds (see Section 2.5).

After a 12-week exposure, the technical surfaces were cleaned using a standardized procedure to remove loose fouling. The purification was carried out with a simple water jet and a conventional GL spraying device (Hornbach, Germany) with a distance of 0.5 m, a height of 0.75 m and an angle of 90° to the technical surfaces at a pressure of 2 bar. The remaining biofouling on the surfaces was documented photographically before and after cleaning and the percentage of fouling was evaluated using the image analysis software ImageJ (National Institutes of Health, Bethesda, MD, USA).

## 3. Results

### 3.1. Size and Shape of the Investigated Drifting Seeds

The investigated seeds show various shapes and sizes (Table 2). They vary between 5.47 ± 0.30 mm (*Calamus erinaceus*) and 250.33 ± 8.76 mm (*Rhizophora mangle*) in length and between 3.53 ± 0.18 mm (*Dypsis paludosa*) and 42.71 ± 2.35 mm (*Entada rheedii*) in width.

### 3.2. Results of Fouling Experiments with Drifting Seeds

After repeated exposure of the 43 species of drifting seeds in the North Sea, 6 species (*Acoelorrhaphe wrightii, Acrocomia totai, Coccothrinax borhidiana, Dypsis paludosa, Erythrina berteroana, Ipomoea alba, Licuala spinosa, Pseudophoenix sargentii and Sapindus saponaria*) showed no fouling, whereas 37 species showed fouling with e. g., barnacles. During several probing campaigns, six drifting seed species showed no fouling under North Sea water conditions.

### 3.3. Results of Surface Analysis of Drifting Seeds with Scanning Electron Microscopy (SEM)

Figure 1 shows SEM images of the surface structure of the drifting seeds species with antifouling properties. At first glance, many of the seeds looked smooth. However, at the micro scale just a few species demonstrated more or less smooth surfaces (*Erythrina berteroana*, *Ipomoea alba*, *Sapindus saponaria*). Even these surfaces revealed surface irregularities up to 5 µm. The surface irregularities of the other seeds are in a range of about 20–50 µm. Their shapes varied between cellular, knotty, flaky and irregular.

Regular honeycomb surface structures are visible structures of *Acoelorrhaphe wrightii* and *Licuala spinosa* in SEM images. *Acoelorrhaphe wrightii* shows structures with a width of 15.21 ± 2.60 µm and a length of 21.76 ± 2.47 µm (10 measurements per 5 surfaces) and *Licuala spinosa* shows structures with a width of 19.08 ± 4.45 µm and a length of 28.07 ± 5.15 µm (10 measurements per 5 surfaces).

### 3.4. Results of the Quantification of the Three Dimensional Surface Roughness of Drifting Seeds

The mean roughness (Ra) of the drifting seeds could be determined by three-dimensional topographic measurements. The results of the drifting seeds are shown in Figure 2. Mean roughness values could be determined in the range from 0.36 ± 0.03 µm (*S. saponaria*) to 2.09 ± 0.46 µm for *L. spinosa* (*n* = 5 per surface with the tilt angles 0°–10°–20°). The values of *E. berteroana* (0.41 ± 0.04 µm) and *I. alba* (0.45 ± 0.02 µm) are in the same range as those for *S. saponaria*. Surfaces from *A. wrightii* (0.88 ± 0.31 µm) and *C. borhidiana* (1.59 ± 0.20 µm) showed similar mean roughness values as the reference glass ball (1.03 ± 0.13 µm).

### 3.5. Reproduction of the Surface Structures on Technical Surfaces

Due to the geometric shapes of the plant seeds, an exact replicate of the entire seed with its natural surfaces was not possible. Alternatives for the reproduction of the surface structures were test surfaces made of 2-component silicone system with integrated particles, which were successfully produced.

Figure 3 shows SEM images of the test surfaces with integrated Poraver and Q-Cel particles compared to a test surface without particles. It is evident that the particles could be successfully integrated into the artificial surfaces, even though the particles are not regularly distributed, have different sizes and partly show a deformation of the spherical shape.

### 3.6. Results of the Fouling Experiments on Technical Surfaces

The technical surfaces were successfully salvaged after an exposure period of 12 weeks each. The cleaned surfaces (see Figure 4) showed a variation in the growth of barnacles. Whereas all the surfaces were almost entirely covered in growth, the biofouling on surfaces with integrated particles did not adhere as strongly during the water jet cleaning.

In addition, on the reference surface made of V2A steel (Figure 4a) the barnacles lie close to each other and occupy almost 100% of the test area. In contrast, the silicone surfaces containing particles show significantly less vegetation. On the surfaces with Poraver particles (Figure 4b), the barnacles are significantly larger and closer to each other compared to the surface with Q-Cels (Figure 4c). On the surface with Q-Cels, hardly any growth of barnacles is visible. Only a few small barnacles have settled.

Figure 5 shows the comparison of biofouling on the studied surfaces. The surface V2A steel was included as reference, its surface covered with growth for almost 100% before (97.99% ± 1.31%) and after cleaning (96.41% ± 1.45%). However, the surfaces with integrated particles show different results: Before cleaning, a high percentage of the surface is covered in biofouling, for Poraver particles 96.47% ± 4.66% and for Q-Cel particles 93.67% ± 3.57%. After cleaning, the surfaces with the particles showed a very significant reduction in fouling. On surfaces with Poraver particles, the growth was reduced to 71.93% ± 7.42% and on surfaces with the Q-Cel particles (34.78% ± 4.31%) the growth was reduced to less than 50%.

## 4. Discussion

Marine organisms exhibit a variety of different surface structures (grooves, dents, spines, scales) and sizes (approximately 1 to 200 µm) with antifouling properties [25]. These are developed so that they solely serve the organism in an optimal way in its ecological niche, successfully repelling epibionts living in its habitat. As the results of the present study show, plant seeds are also capable of effectively repulsing epibionts. In this study, 6 out of the 43 plant seed species exhibited growth-minimizing properties during repeated exposures of 12 weeks each.

While the sizes of all examined seeds were in the range from 5.47 to 250.33 mm in length and from 3.53 to 42.71 mm in width, the successful identified 6 species belong to the smaller sized of the seeds (lengths: 6.84–11.61 mm; width: 5.38–10.63 mm). Large plant seeds provide more attachment points for biofouling in general than small ones, which may result in better growth on them [26]. So it may be not surprising that the small plant seed species remain free of barnacles settlement.

Next to the size and shape of the drifting seeds, the microstructured surface properties can be regarded as the major reason for the minimization of fouling such as the regular honeycomb surfaces of *A. wrightii* and *L. spinosa*. Research has already verified the influence of surface microstructures in the defense against fouling. Studies of pilot whale and shark skins, mussel shells and crab housings have explored their anti-adhering surface properties [20,21,27]. Here, the natural antifouling defense is lost as soon as the surfaces are destroyed due to age, injury or disease. However, the efficacy of microtopography is in many cases also dependent on the size of the microtopography in proportion to the size of the settling propagule or larvae [28]. This dependence can be excluded for the investigated drifting seed species: Seeds like *C. erinaceus* (5.47 ± 0.30 mm length) or *C. cujete* (7.36 ± 0.68 mm length) showed fouling although they were of similar size as the 6 identified seed species that exhibited antifouling microstructures.

Three dimensional topographical measurements using MEX software are widely used in various research areas such as forensics, anthropology, archaeology, dermatology and implantology [29,30]. This software also provides a good potential for characterizing and comparing the roughness parameters of the surface properties of drifting seeds. It also allows a quantitative evaluation of data and parameters that can be compared with ISO standards [29,30]. With this purely optical measuring method, it was possible to carry out a comparative series of tests to determine the mean roughness parameters of the drifting seeds. A comparison of the values showed that small seeds such as *A. wrightii* (6.87 ± 0.49 mm length), *C. borhidiana* (5.81 ± 0.55 mm length) and *L. spinosa* (6.84 ± 0.67 mm length) as well as the reference glass ball (5 mm in diameter) had higher mean roughness values than larger seeds such as *E. berteroana* (11.35 ± 0.52 mm length), *I. alba* (9.95 ± 0.66 mm length) and *S. saponaria* (11.48 ± 0.55 mm length). Furthermore, the comparison showed that the standard deviations of mean roughness values are larger for smaller than for larger drifting seeds. The mean roughness values determined by this measurement method cannot be compared with other methods because they are different techniques. Studies of mussel surfaces showed very detailed topography parameters such as mean waviness, skewness in the roughness profile, waviness profile, texture aspect ratio or fractal dimension when using laser scanning confocal microcopy (LSCM) [28]. Before confining the measurement methods to be used, white light interferometry, tactile touch measurements and atomic force microscopy (AFM) were also taken into consideration for a more comprehensive and detailed topography measurement of the seeds. However, the seeds themselves were not suitable to be prepared for analysis with those instruments or for measuring those characteristics due to their hardness, microstructures and curvature.

In addition to the physical properties of the organisms, the chemical ingredients or a combination of physical and chemical properties may also play an essential role in the defense against fouling. This study excluded the influence of chemical substances that are part of the plant seeds by only examining the microstructured properties and transferring those to technical surfaces. At first glance, a simple and effective method for taking a molding of the seeds (inclusive size, shape and microstructures) as a whole seemed impossible. That is why a simple method of producing test surfaces was employed using a silicone mixture with different integrated particles that was based on the microstructures of the seeds. Thus at least the microstructured properties of the drifting seeds could be successfully transferred to flat surfaces of 15 × 15 cm for simplification. The silicone mixture was selected for producing the technical surfaces, since silicone has been used for the coatings of underwater ships since the 1970s [31]. Because of their material properties, they already provided an environmentally friendly alternative to toxic TBT-coatings at that time. The mixture of the 2-component silicone system used for producing the technical surfaces is biocide-free, scentless, tasteless, heat-resistant, water-repellent and non-toxic to nature. Due to the quick and easy processing time, the material was very suitable for the production of technical surfaces for this study. In addition, the curing took place almost shrink-free (according to the manufacturer < 0.1% after 7 days).

The surface analyses using SEM clearly showed that the particles were integrated into the silicone and that the replication of the microstructured surfaces would be possible. However, the results show that an even distribution of the particles was difficult to achieve in the respective surfaces due to the manufacturing technique (rolling out of the coating). A more detailed analysis of the particles under the SEM also showed that the corresponding structural shapes and sizes of the particles varied greatly due to their manufacturing method, which prevents uniformity in the molding of the structured surface. In addition, there may have been spots which were covered in silicone but did not contain the integrated particles on the surface so that this may have had a negative effect on the topography. Due to their material composition, it is also possible that some particles in the silicone mixture sank downwards and could hardly affect the formation of the structural surfaces. In addition, the method used does not guarantee an even layer thickness. For example, the soft lithographic technique may be an alternative for taking impression of biological surfaces [32].

The fouling analysis that took place after the retrieval from the study site clearly showed that the areas with a specific structure show a minimization in fouling. Accordingly, the structure of these test surfaces has an influence on the defense of the epibionts. Shortly after the immediate salvage of the test surfaces, all the plates examined showed almost 100% biofouling. However, a significant reduction in the growth of barnacles on the surfaces with the Poraver particles and Q-Cel was observed after cleaning. The comparison of the biofouling values of the cleaned surfaces showed less fouling on the structured surfaces than on the smooth reference surface made of V2A steel: 25.4% for Poraver particles and 63.92% for Q-Cel particles. In this study, the influence of the silicone material itself on the minimization of biofouling can be excluded due to the lack of minimum inflow and minimum activity [31]. These are necessary for the efficacy of the material against the growth. The tidal changes and the tidal range of 3.20 m at the storage site are also not high enough for assuring the minimum inflow or activity. Since the materials used can themselves be excluded to prevent fouling, the produced microstructures of the technical surfaces can be regarded as the cause. The sizes of the particles, their material properties as well as the weight proportions of the respective particles in the test surface certainly play a decisive role. The exact effect needs to be investigated in further studies.

Recent results show that the replication technique based on the work of Koch et al. [33] offers a promising alternative for the transfer of drifting seeds to technical surfaces (Clasen et al., in progress). Surface replication is a simple, cost-effective, efficient and reproducible method for reproducing topographically complex structures [34]. With this technique, natural surfaces can be exactly reproduced, which is why it is already widely used in implant dentistry. The materials used for the impression can vary depending on the field of application. The replication of the microstructural surfaces of the repulsive seeds proved the applicability of this method.

## 5. Conclusions

The present study confirms that not only marine organisms but also terrestrial plant seeds serve as a model for non-toxic antifouling solutions.

It could be shown that the geometric shape and the surface microstructures of the drifting seeds have a considerable influence on the growth defense. First indications suggest that the curvature may also play a factor in minimizing the fouling effects.

In the case of organic materials, the interplay of different growth-minimizing properties such as chemical and physical characteristics cannot be ruled out. It is therefore obvious that, in addition to the physical properties of the drifting seed species, the chemical ingredients such as the natural bioactive products of terrestrial plants [35] or a combination of these mechanisms may also play a role in defending against biofouling. For determining the exact mechanisms, further analyses are necessary which should focus on the effect of individual properties in order to be able to exclude or to confirm correlations between physical and chemical mechanisms.

## Figures and Tables

**Figure 1 biomimetics-04-00037-f001:**
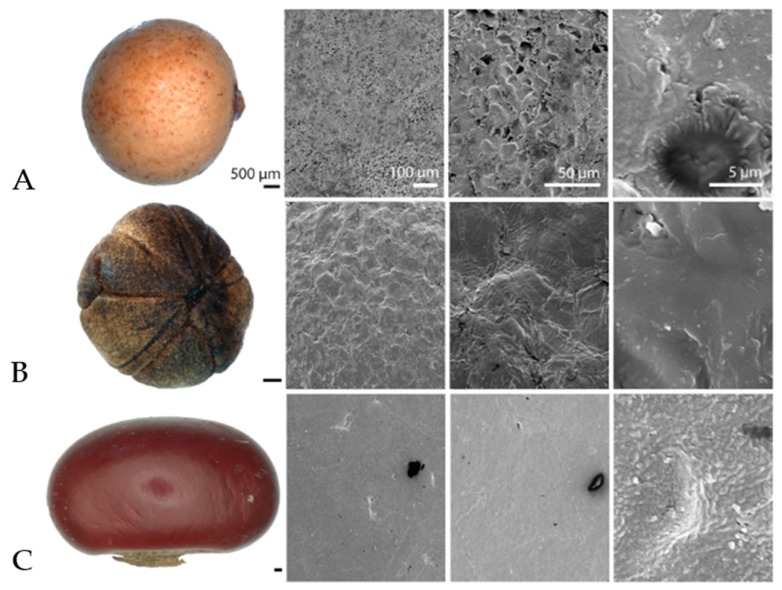
Samples and Scanning Electron Microscopy (SEM) images of seeds with antifouling properties (**A**) *Acoelorrhaphe wrightii*, (**B**) *Coccothrinax borhidiana*, (**C**) *Erythrina berteroana*, (**D**) *Ipomoea alba*, (**E**) *Licuala spinosa* and (**F**) *Sapindus saponaria*.

**Figure 2 biomimetics-04-00037-f002:**
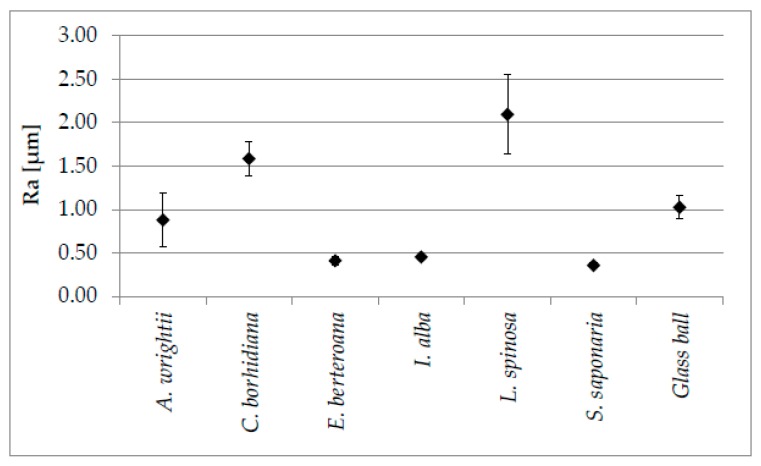
Mean roughness (Ra) of the profiles of six drifting seed species and their standard deviations (*n* = 5) in comparison to the reference glass ball determined by means of the digital elevation models. (Standard deviation of *E. berteroana, I. alba* and *S. saponaria* are not visible; <9% of the mean).

**Figure 3 biomimetics-04-00037-f003:**
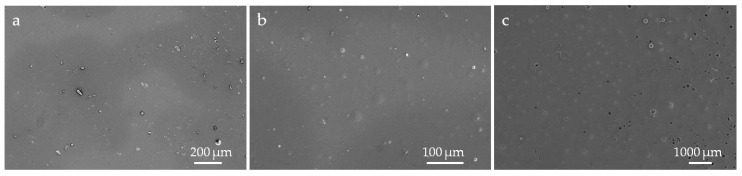
SEM images of test surfaces with integrated particles. From left to right: (**a**) surface of the silicone mixture without particles, (**b**) surface of the silicone mixture integrated with Poraver particles and (**c**) surface of the silicone mixture integrated with Q-Cel particles.

**Figure 4 biomimetics-04-00037-f004:**
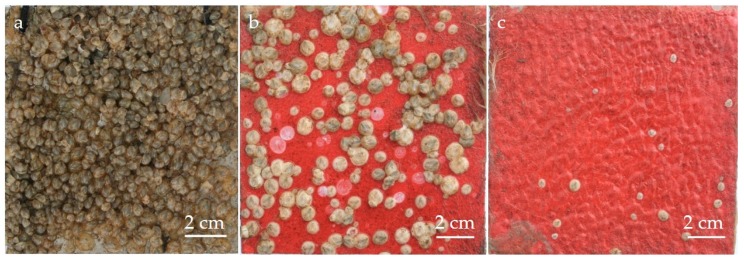
Adhesion of barnacles to surfaces after 12 weeks of exposure to the North Sea (after cleaning). (**a**) V2A-steel reference, (**b**) silicone mixture with Poraver particles and (**c**) silicone mixture with Q-Cel.

**Figure 5 biomimetics-04-00037-f005:**
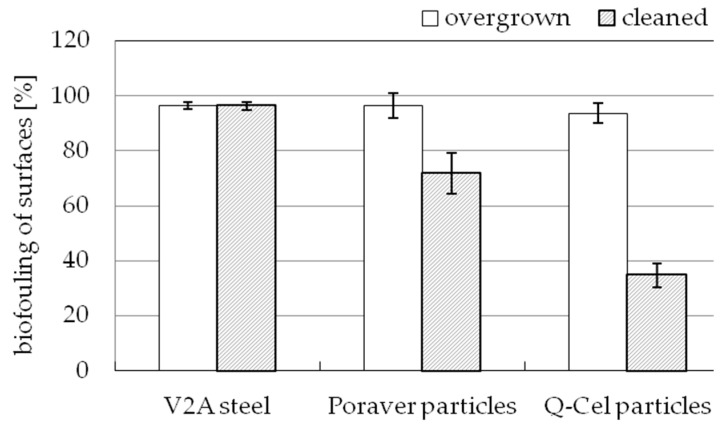
Mean values and standard deviations of the percentage biofouling (*n* = 3) of the test surfaces removed compared to the reference V2A-steel before and after water jet cleaning.

**Table 1 biomimetics-04-00037-t001:** List of selected drifting seeds used in the study. The table shows the plants families, the species names and the places of origin of the respective drifting seeds.

Family	Species	Regions and Countries of Origin
Anacardiaceae	*Spondias mombin*	Peru, Brazil, Venezuela, Bolivia, Colombia, Mexico, Belize, Costa Rica, West Indies
Annonaceae	*Annona glabra*	Florida, Caribbean, Central America, South America, West Africa
Apocynaceae	*Thevetia peruviana yellow*	Mexico, Central America, South America
Arecaceae	*Acoelorrhaphe wrightii*	West Indies, Cuba, Central America
	*Acrocomia totai*	Brazil, Bolivia, Argentina
	*Allagoptera arenaria*	Brazil
	*Archontophoenix myolensis*	Queensland
	*Astrocaryum standleyanum*	South America, Central America, Ecuador to Panama, Colombia to Ecuador
	*Astrocaryum vulgare*	Brazil, Peru, Venezuela
	*Calamus erinaceus*	Southeast Asian, Southern Thailand, Philippines, Malaysia, Brunei, Singapore, Indonesia
	*Coccothrinax borhidiana*	Cuba
	*Coccothrinax boschiana*	Dominican Republic
	*Dypsis paludosa*	Madagascar
	*Dypsis rivularis*	Madagascar
	*Elaeis guineensis*	Africa, tropical America, Southeast Asia
	*Licuala paludosa*	Southeast Asia
	*Licuala spinosa*	Southeast Asia
	*Manicaria saccifera*	Central America, South America
	*Orbignya cohune*	Central America, Mexico, Costa Rica
	*Phoenix roebelenii*	Southeast Asia
	*Pritchardia minor*	Hawaii
	*Pseudophoenix sargentii*	Mexico, Belize, Northern Caribbean
	*Raphia australis*	Tropics, Africa, Madagascar
Bignoniaceae	*Crescentia cujete*	West Indies, South-Mexico to Peru, Brazil, Caribbean
Bombacaceae	*Adansonia digitata*	Africa, Madagascar, Sri Lanka
Combretaceae	*Laguncularia racemosa*	West Africa, North America, South America, Mexico to Brazil, Ecuador
	*Terminalia catappa*	Southeast Asia, West Africa
Convolvulaceae	*Ipomoea alba*	Argentina, Mexico, Florida
	*Ipomoea pes-caprae*	Tropics
	*Merremia tuberosa*	Central America, South America, Mexico, Tropics, Subtropics
Euphorbiaceae	*Ricinus communis*	Northeast Africa, Tropics
Fabaceae	*Canavalia rosea*	Tropics, Subtropics, Florida, California, Texas, Mexico
	*Entada rheedii*	Africa, Southeast Asia, India to China, Philippines, Northern Australia
	*Erythrina berteroana*	Central America, South America
	*Mucuna nigricans*	Asia, Florida
	*Mucuna sempervirens*	Bhutan, China, Japan
	*Mucuna urens*	Tropics
	*Sophora tometosa*	South Florida, Cuba, Caribbean
Juglandaceae	*Carya illionensis*	Mexico
Nelumbonaceae	*Nelumbo nucifera*	Asia, China, Japan, India
Pandanaceae	*Pandanus conicus*	Australia
	*Pandanus monotheca*	Thailand
	*Pandanus pulcher*	Madagascar
	*Pandanus utilis*	Tropics, Madagascar
Polygonaceae	*Coccoloba uvifera*	Tropics, tropical America, Caribbean, South America
Rhizophoraceae	*Rhizophora mangle*	Tropics, West Africa, North America, South America
Sapindaceae	*Sapindus saponaria*	South Carolina, Caribbean, Central America, South America

**Table 2 biomimetics-04-00037-t002:** List of morphological data and geometric shapes of the drifting seeds.

Geometric Shape	Species of Drifting Seeds	*n*	Arithmetic Mean and Standard Deviation of Length (mm)	Arithmetic Mean and Standard Deviation of Width (mm)
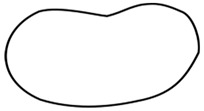 bean	*Adansonia digitata*	20	12.13 ± 0.49	9.58 ± 0.55
	*Canavalia rosea*	20	13.72 ± 0.80	9.20 ± 0.99
	*Entada rheedii*	04	46.50 ± 3.83	42.71 ± 2.35
	*Erythrina berteroana*	20	11.35 ± 0.52	7.63 ± 0.34
	*Mucuna nigricans*	20	23.33 ± 0.90	19.11 ± 2.33
	*Mucuna semperviren*	20	30.27 ± 1.53	19.50 ± 1.60
	*Mucuna urens*	10	27.77 ± 2.40	22.87 ± 2.48
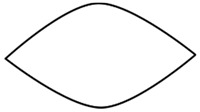 nut	*Allagoptera arenaria*	22	14.09 ± 4.42	10.02 ± 1.15
	*Archontophoenix myolensis*	44	12.76 ± 0.87	8.40 ± 0.28
	*Astrocaryum standleyum*	11	31.00 ± 3.77	23.37 ± 1.38
	*Astrocaryum vulgare*	12	27.50 ± 1.65	23.09 ± 1.00
	*Carya illinoensis*	20	31.24 ± 3.99	15.24 ± 0.87
	*Coccoloba uvifera*	20	11.61 ± 0.95	8.40 ± 0.47
	*Dypsis paludosa*	38	8.71 ± 0.71	3.53 ± 0.18
	*Dypsis rivularis*	44	6.77 ± 0.69	4.28 ± 0.35
	*Elais guineensis*	11	22.15 ± 5.37	18.27 ± 2.36
	*Laguncularia racemosa*	20	14.88 ± 2.11	6.35 ± 0.75
	*Licuala paludosa*	23	6.79 ± 0.49	5.99 ± 0.19
	*Licuala spinosa*	20	6.84 ± 0.67	5.38 ± 0.27
	*Orbignya cohune*	8	60.00 ± 2.62	34.58 ± 1.51
	*Pritchardia minor (P. eriophora)*	25	16.33 ± 0.85	12.09 ± 0.74
	*Raphia australis*	21	67.00 ± 5.20	37.02 ± 2.20
	*Ricinus communis*	30	10.93 ± 0.52	7.07 ± 0.32
	*Terminalia catappa*	20	56.45 ± 6.03	35.08 ± 2.86
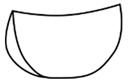 boat	*Ipomoea alba*	30	9.95 ± 0.60	7.53 ± 0.57
	*Ipomoea pes-caprae*	30	9.38 ± 0.56	7.05 ± 0.41
	*Merremia tuberosa*	20	17.00 ± 1.54	14.24 ± 0.91
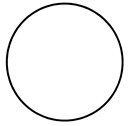 spherical	*Acoelorrhaphe wrightii*	46	6.87 ± 0.49	6.69 ± 0.45
	*Acrocomia totai*	10	21.91 ± 2.13	20.73 ± 1.39
	*Coccothrinax borhidiana*	20	5.81 ± 0.55	5.73 ± 0.59
	*Coccothrinax boschiana*	23	11.61 ± 0.95	8.40 ± 0.47
	*Manicaria saccifera*	21	37.37 ± 2.26	37.23 ± 2.20
	*Nelumbo nucifera*	20	14.21 ± 0.53	11.26 ± 0.45
	*Pseudophoenix sargentii*	34	9.72 ± 0.84	9.19 ± 0.07
	*Sapindus saponaria*	20	11.48 ± 0.55	10.63 ± 0.57
other shapes	*Annona glabra*	20	13.97 ± 1.02	8.71 ± 0.75
	*Calamus erinaceus*	45	5.47 ± 0.30	5.30 ± 0.38
	*Crescentia cujete*	20	7.36 ± 0.68	6.68 ± 0.69
	*Pandanus conicus*	10	51.33 ± 5.70	40.00 ± 5.86
	*Pandanus monotheca*	10	32.26 ± 2.85	10.21 ± 1.57
	*Pandanus pulcher*	10	21.02 ± 2.10	5.27 ± 1.06
	*Pandanus utilis*	10	34.46 ± 4.62	26.10 ± 10.00
	*Phoenix roebelenii*	79	8.55 ± 1.18	4.35 ± 0.44
	*Rhizophora mangle*	6	250.33 ± 8.76	14.01 ± 0.64
	*Spondias mombin*	10	22.44 ± 2.01	14.17 ± 0.92
	*Thevetia peruviana yellow*	20	33.91 ± 3.40	18.41 ± 1.39

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
