# Peer review of "Microstructural Surface Properties of Drifting Seeds—A Model for Non-Toxic Antifouling Solutions"

_biomimetics, 2019, doi:10.3390/biomimetics4020037_

Round 1
Reviewer 1 Report
I think the topic of the paper has value to the community of scientists interested in biofouling. The authors choose a matrix that has potentially interesting surface characteristics for antifouling.
I think that the introduction deals too much with the antifouling problem, which has been very well addressed at this point, rather than addressing the reference to textures and biomimetic solutions. Some of course have been included, but an expanded introduction focusing on texture rather than the problem would be advisable.
The manuscript could be published if the authors were to address a degree of improvement. It is not clear what the value of the silicone chemistry or the specific texture offers. There should be a clear discussion on both in the context of attachment. Regarding the chemistry, it has been previously shown and perhaps the authors could comment on the composition and ratio in relation to replication issues observed.
With regard to the texture Authors have selected a large number of sample types. It would be more beneficial to the study if there was a smaller selection of target seeds with different textures that were fully characterised. By this I mean also, that the attachment theory should be considered which will also need reference to dimensions within the textures. What I like about this work, is the potential for replication of such simple textures and therefore the value within the antifouling industry. However, it will be necessary to understand better why certain structures facilitate lighter attachment.
Author Response
Dear Rewiever,
I would like to thank you for your advice and revision requests for my paper.
In consultation with the 2nd author, I have worked on all points that the review mentioned and proposed and would like to submit the revised version of the paper. The revised text passages are presented in orange for a better overview.
This article is intended to be the first study in a series of studies to be published in the near future. Further articles are already in progress. The next article, for example, will focus on the replication method.
Here are my replies to your comments:
You wrote:
I think that the introduction deals too much with the antifouling problem, which has been very well addressed at this point, rather than addressing the reference to textures and biomimetic solutions. Some of course have been included, but an expanded introduction focusing on texture rather than the problem would be advisable.
--> The introduction deals strongly with the problem of antifouling. Since further articles concerning antifouling in plant seeds are planned or in progress, this article should be seen as a complete introduction to the problem.
--> In nature there is a multitude of biological models. Due to these, only a few of them were discussed in the article (see lines 73-88).
You continued to write:
It is not clear what the value of the silicone chemistry or the specific texture offers. There should be a clear discussion on both in the context of attachment. Regarding the chemistry, it has been previously shown and perhaps the authors could comment on the composition and ratio in relation to replication issues observed.
-->The use of the material silicone was included in the discussion as recommended (see lines 319-340).
You also wrote:
With regard to the texture Authors have selected a large number of sample types. It would be more beneficial to the study if there was a smaller selection of target seeds with different textures that were fully characterized. By this I mean also, that the attachment theory should be considered which will also need reference to dimensions within the textures. What I like about this work, is the potential for replication of such simple textures and therefore the value within the antifouling industry. However, it will be necessary to understand better why certain structures facilitate lighter attachment.
--> The study did indeed examine many different plant seeds. However, this was necessary because no studies have yet been carried out by other researchers and documented in the literature. The results show that only a few seeds have antifouling effects. So only these seeds have been the focus of detailed analyses.
I hope that I have heeded all the requests for changes and that I will receive a positive response from you.
I remain with cordial greetings,
Antje Clasen
Reviewer 2 Report
In the main, it appear to be two separate studies merged together, neither of which appear complete in themselves: the “seed” aspect and the “inspiration” for the “technical surfaces”. A complete (or at least further) characterisation of the surface structure of the subset of seeds that appear to have some biofouling reduction would have been very useful, indeed there are many possible reasons other than surface topography of the seeds that could affect AF performance shown in this study (perhaps sloughing of surface materials, swelling in seawater, leaching of antimicrobials or other secondary metabolites from “waxy” surface coverings, materials stiffness and “compliance”, hydrophobicity etc. etc.) – I think perhaps the authors imply too readily that the antifouling effects seen are due to surface topography, without due discussion of the (at times quite significant) limitations of this aspect of their study. Indeed – even the surface structure is not well characterised in this study – for surface characterisation see for example Scardino, A. J., et al. "Biomimetic characterisation of key surface parameters for the development of fouling resistant materials." Biofouling 25.1 (2009): 83-93.
It appears to me that the second study consists of the “technical surfaces” and the fouling of those surfaces – again to be of more use to in the development of antifouling materials/surface, I feel that these technical surfaces/materials would need to characterised in much greater detail – surface energy/wettability/the various moduli etc. in order that results presented be in any way reproducible.
Was the water jet calibrated for fouling removal? - if so details - why is the control fouling reported as essentially the same before and after cleaning?
Finally, the discussion of the limitations of the surface topography approach to AF surfaces could be improved – see for example: Sullivan, T., and F. Regan. "Marine diatom settlement on microtextured materials in static field trials." Journal of materials science 52.10 (2017): 5846-5856.
Some minor points in general: there are quite a few grammatical errors in this – in general the paper needs to be proof read thoroughly.
Some of the terminology and structure of the paper is confusing i.e. the creation of “technical surfaces” – perhaps artificial surface/material would be clearer. – use of “Chapters” etc...
Author Response
Dear Rewiever,
I would like to thank you for your advice and revision requests for my paper.
In consultation with the 2nd author, I have worked on all points that the review mentioned and proposed and would like to submit the revised version of the paper. The revised text passages are presented in orange for a better overview.
This article is intended to be the first study in a series of studies to be published in the near future. Further articles are already in progress. The next article, for example, will focus on the replication method.
Here are my replies to your comments:
You wrote:
In the main, it appear to be two separate studies merged together, neither of which appear complete in themselves: the "seed" aspect and the "inspiration" for the "technical surfaces".
--> No, these are not 2 separate studies. This study is mainly concerned with the investigation of the antifouling effect of the selected plant seeds and their transferability to a technical surface. The transfer to a technical surface is necessary, since in biological systems, in addition to the surface properties, chemistry can also have an influence on the minimization of fouling (see lines 97-102 or 311-319).
You continued to write:
A complete (or at least further) characterisation of the surface structure of the subset of seeds that appear to have some biofouling reduction would have been very useful. .....Indeed - even the surface structure is not well characterised in this study - for surface characterisation see for example Scardino, A. J., et al. "Biomimetic characterisation of key surface parameters for the development of fouling resistant materials." Biofouling 25.1 (2009): 83-93.
--> For the characterization of the surface structures of the seeds, sections on roughness measurement were included (see chapters 2.3 and 3.4) and discussed (see lines 288-310).
--> In addition, the recommended article by Scardino et al. is cited.
You also wrote:
It appears to me that the second study consists of the “technical surfaces” and the fouling of those surfaces – again to be of more use to in the development of antifouling materials/surface, I feel that these technical surfaces/materials would need to characterised in much greater detail – surface energy/wettability/the various moduli etc. in order that results presented be in any way reproducible.
--> The production of the technical surfaces shows only the beginnings of further investigations mentioned above. The focus of this article is on the production by means of replication.
You wrote:
Was the water jet calibrated for fouling removal? - if so details - why is the control fouling reported as essentially the same before and after cleaning?
--> Now, the cleaning procedure is described in lines 179-182.
--> The results in Fig. 5 show that the reference V2A steel has a growth of nearly 100% (see line 257-259) before and after the cleaning compared to the other technical surfaces. The results were taken up and discussed in lines 344-356.
You continued to write:
Finally, the discussion of the limitations of the surface topography approach to AF surfaces could be improved – see for example: Sullivan, T., and F. Regan. "Marine diatom settlement on microtextured materials in static field trials." Journal of materials science 52.10 (2017): 5846-5856.
--> The mentioned article was very interesting and brought more ideas to light. The content was picked up and used in the discussion (see lines 271-275).
I hope that I have heeded all the requests for changes and that I will receive a positive response from you.
I remain with cordial greetings,
Antje Clasen
Round 2
Reviewer 1 Report
Overall the modifications to the paper have resulted in a very nice study. M y only remaining concern is that the conclusion is weak. The conclusion would not draw reader to novel outcomes of the research. The authors must address this in my view before accepting the paper. I would suggest that the conclusion links with the original hypothesis and aims of the study and clear statements of evident, success or otherwise must be given.
Author Response
Dear Reviewer,
I would like to thank you again for your advice and revision requests for my paper.
In consultation with the 2nd author, I have mainly worked on the conclusion and would like to submit the revised version of the paper. The revised text passages are highlighted in orange.
I hope that I have heeded all the requests for changes and that I will receive a positive response from you.
I remain with cordial greetings,
Antje Clasen

Reviewer 2 Report
The revised manuscript is very much improved - the objectives and rationale for the study are much clearer - as is the discussion of limitations of the work and the significance.
However, I'm confused by the argument/ justification for methods used and explanation that surface topography could not be characterised: lines 303-310, Bivalve shells are also curved - perhaps the meaning has been lost here or is unclear?
I would also consider re-writing the conclusions - they don't seem to relate well to your results and are somewhat unclear overall.
Author Response
Dear Reviewer,
I would like to thank you again for your advice and revision requests for my paper.
In consultation with the 2nd author, I have worked on lines 303-310 and the conclusion. I would like to submit the revised version of the paper. The revised text passages are highlighted in orange.
I hope that I have heeded all the requests for changes and that I will receive a positive response from you.
I remain with cordial greetings,
Antje Clasen
